# Molecular Differences Based on Erythrocyte Fatty Acid Profile to Personalize Dietary Strategies between Adults and Children with Obesity

**DOI:** 10.3390/metabo11010043

**Published:** 2021-01-08

**Authors:** Iker Jauregibeitia, Kevin Portune, Sonia Gaztambide, Itxaso Rica, Itziar Tueros, Olaia Velasco, Gema Grau, Alicia Martín, Luis Castaño, Anna Vita Larocca, Federica Di Nolfo, Carla Ferreri, Sara Arranz

**Affiliations:** 1AZTI, Food Research, Basque Research and Technology Alliance (BRTA), Parque Tecnológico de Bizkaia, Astondo Bidea, Edificio 609, 48160 Derio–Bizkaia, Spain; ijauregibeitia@azti.es (I.J.); kportune@azti.es (K.P.); itueros@azti.es (I.T.); 2Biocruces Bizkaia Health Research Institute, Cruces University Hospital, CIBERDEM/CIBERER, UPV/EHU, Endo–ERN, 48903 Barakaldo, Spain; MARIASONIA.GAZTAMBIDESAENZ@osakidetza.eus (S.G.); itxaso.ricaechevarria@osakidetza.eus (I.R.); olaia.velascovielba@osakidetza.eus (O.V.); MARIAGEMA.GRAUBOLADO@osakidetza.eus (G.G.); Alicia.MartinNieto@osakidetza.eus (A.M.); lcastano@osakidetza.eus (L.C.); 3Lipidomic Laboratory, Lipinutragen srl, Via di Corticella 181/4, 40128 Bologna, Italy; annavita.larocca@gmail.com (A.V.L.); federica.dinolfo@lipinutragen.it (F.D.N.); 4Consiglio Nazionale delle Ricerche, ISOF, Via Piero Gobetti 101, 40129 Bologna, Italy

**Keywords:** lipid metabolism, mature erythrocyte, obesity, precision nutrition

## Abstract

As the obesity epidemic continues to grow inexorably worldwide, the need to develop effective strategies to prevent and control obesity seems crucial. The use of molecular tools can be useful to characterize different obesity phenotypes to provide more precise nutritional recommendations. This study aimed to determine the fatty acid (FA) profile of red blood cell (RBC) membranes, together with the evaluation of their dietary intake and biochemical parameters, of children and adults with obesity. An observational study was carried out on 196 children (113 with normal weight and 83 with obesity) and 91 adults (30 with normal weight and 61 with obesity). Mature RBC membrane phospholipids were analyzed for FA composition by gas chromatography-mass spectrometry (GC-MS). Dietary habits were evaluated using validated food frequency questionnaires (FFQ). Children with obesity presented higher levels of ω-6 polyunsaturated FAs (mainly linoleic acid, *p* = 0.01) and lower values of ω-3 FAs (mainly DHA, *p* < 0.001) compared with adults. Regarding blood biochemical parameters, children with obesity presented lower levels of glucose, LDL cholesterol, and alanine aminotransferase compared with adults with obesity. These lipidomic differences could be considered to provide specific nutritional recommendations for different age groups, based on an adequate fat intake.

## 1. Introduction

In the last decades, unhealthy dietary patterns are increasing and affecting the prevalence of noncommunicable diseases in the world. According to the World Health Organization (WHO), worldwide obesity has nearly tripled since 1975, indicating that, in 2016, 39% of adults aged 18 years and over, were overweight and 13% suffered from obesity [1]. In addition to affecting the adult population, obesity is becoming a rising problem affecting children and adolescents as well. As the WHO states, more than 340 million children and adolescents, around one in three from 5 to 19 years, were overweight or obese in 2016 and 38 million children under the age of 5 were overweight or obese in 2019 [1].

Obesity prevention and treatment strategies include lifestyle and behavioral interventions, focused on changes in diet and physical activity. Low-fat diets, aimed at reducing caloric intake, have been the most recommended strategy for people with obesity in the past decades [2]. Moreover, there is not enough evidence from randomized control trials supporting the beneficial effects of low-fat diets over other dietary interventions for long–term weight loss [2]. Besides, recent scientific evidence showed that low-fat diets reduced LDL and HDL cholesterol and increase triglycerides. Further, the replacement of saturated fatty acids (SFA) with monounsaturated fatty acids (MUFA) has been proposed as an appropriate strategy to reduce obesity, since substituting SFA with MUFAs raises HDL-cholesterol levels, improves insulin sensitivity, and lowers LDL-cholesterol levels [3,4,5]. Other dietary plans, as a strategy for obesity management, have been proposed including low–carbohydrate diets, high-protein diets, very-low-calorie diets with meal replacements, Mediterranean diet, and diets with intermittent energy restrictions, evidencing that a successful diet to reduce weight must be healthy, balanced and without nutritional deficiencies. In any case, most of them include general dietary recommendations rather than specific dietary plans based on individual metabolism [6].

However, obesity prevalence in both child and adult populations continues increasing worldwide, suggesting that, personalized intervention strategies could provide precise nutritional guidance and contribute to successful long-term interventions [7]. Even though dietary guidelines for macronutrients intake in adults and children are established [8,9], according to the different requirements of both population groups, especially from energy intake, interventions to control obesity in children and adults are not specific nor differentiated, regarding the intake of food groups or specific nutrients. For that reason, the optimal macronutrient distribution of the diet to improve weight status is unclear [10]. The use of molecular tools (metabolomic, nutrigenetic, metagenomic, etc.) can provide new scientific evidence related to the characterization of different obesity phenotypes together with the impact of diet on metabolism [11]. This can be useful to personalize therapy and contribute to providing more precise nutritional recommendations, mainly for an adequate fat intake for different age groups and health conditions [12,13,14].

The use of mature erythrocyte membrane as a representative site for all other body tissues in FA profiling is an established protocol for membrane-based molecular diagnostics [15,16,17]. The measure of the lipid profile at the cellular level, precisely at the membrane phospholipid level, provides not only information related to the nutritional status of an individual, but also information related to FA metabolism that is involved in the formation of the most important lipid building blocks for cell life, which are the phospholipids. This approach has a profound diagnostic meaning, not only from the biochemical point of view related to the lipid pathways, but also from the biophysical and biological consequences, since the balance reached by the FA components of the membrane phospholipids must respect the tissue type and, ultimately, satisfy the homeostatic requirement for the optimal cell functioning [18].

This study aimed to evaluate lipid profile differences in mature RBC membranes between children and adults with obesity, in relation to their nutrient intake. Defining these differences in RBC FA profiles, related to individual molecular and nutritional status, will allow the design of differentiated nutrition strategies for children and adults with obesity, giving relevance to the functional roles of different types of dietary fats.

## 2. Results

### 2.1. Descriptive Characteristics of the Participants

A total of 83 children with obesity (26 boys and 57 girls) between 6 to 16 years old and a group of 61 adults with obesity (19 males and 42 females) between 19 to 68 years old participated in the study (Appendix A). At the same time, control subjects, consisting of 113 children and 30 adults with normal weight and same age ranges were also included. A matched gender distribution was found for children and adults with obesity (*p* = 0.83) but for normal weight group, there was not a matched gender distribution (*p* = 0.04). Obvious differences were observed for age between children and adults with obesity and children and adults with normal weight.

### 2.2. Red Blood Cell Membrane Fatty Acids Profile

The pediatric group with obesity showed lower levels of palmitic acid and cis–vaccenic acid (*p* = 0.01 and *p* = 0.05 respectively) compared to the adult group with obesity (Table 1). LA, DGLA, and total ω–6 FA levels in the pediatric group with obesity were higher compared with the adult group with obesity (*p* = 0.01, *p* = 0.04, and *p* < 0.01 respectively). DHA and total ω-3 FA levels were lower for the pediatric group with obesity (*p* < 0.01 and *p* < 0.01 respectively) and hence, the ω-6/ω-3 ratio was higher (*p* < 0.01). Regarding other indexes, PUFA balance, PI, and UI were lower for the pediatric group with obesity (*p* < 0.01, *p* = 0.01 and *p* = 0.04 respectively).

After observing these differences, a sample of normoweight adults (30) and children (113) were analyzed to determine if these differences observed between the groups with obesity were due only to age differences or whether they could be attributed to metabolic differences. Similar patterns for LA, DGLA, DHA, total ω-3 FA levels, total ω–6 FA levels, and ω-6/ω-3 ratios were observed in the normoweight populations, but no differences for palmitic acid, cis–vaccenic acid, or UI were observed when comparing adult and child populations with normal weight. The DNL index [19], which is the ratio between C16:0/C18:2 ω-6 fatty acids, that correlates directly with the liver fat content, appears in higher levels for adults with normal weight and obesity when compared with the respective child populations (*p* < 0.01 for both).

### 2.3. Blood Biochemical Parameters

Blood biochemical parameters were determined in a subsample of the study (Table 2). Glucose levels were significantly higher in the adult populations with obesity (*p* < 0.001) but not for the groups with normal weight (*p* = 0.38). Alanine Aminotransferase (ALT/GPT) values were lower for children with obesity compared with adults with obesity (*p* = 0.03) but no differences were observed between groups with normal weight. Cholesterol and triglycerides were statistically higher for adults with obesity (*p* < 0.01 for both) compared to children with obesity, but no differences between groups with normal weight were observed.

### 2.4. Food Groups

To compare the dietary pattern of each study group, we considered food groups as shown in Appendix A and observed that both adults with obesity and with normal weight showed a higher intake of vegetables, olive oil, white meat, oily fish, sugary drinks and dried fruits and nuts (*p* < 0.01 for all) compared to a group of children. On the other hand, cereals, legumes, and juice intakes were higher for the pediatric population (*p* < 0.01 for both obese and normoweight). In any case, these results should be taken into account from the perspective that, adults, reported a higher intake of daily calories, so when comparing food groups in grams per day units, it is normal to observe differences.

### 2.5. Nutrient Intake

Regarding macronutrient intake shown in Table 3, as both population groups differ in terms of quantity requirements, to compare each other, variables were expressed in % of the energy obtained from each macro-micronutrient. Differences between obese adults and children were observed for total calories (Kcal/day) and the intake of carbohydrates, simple sugars, and total lipids. Some of these differences, such as calorie differences (*p* < 0.01) can be related to different requirements depending on age. The distribution of the energy intake obtained from macronutrients differs between both populations for carbohydrates (*p* < 0.01), being higher for the pediatric population. Although adults showed, proportionally, a higher intake of total lipids than children (*p* < 0.01), the pediatric population showed a higher intake of stearic acid (*p* = 0.04). Oleic acid and total MUFA intake were higher for the adult population. Regarding PUFAs, total ω-6, corresponding ω-6 fatty acids (LA and AA), and total ω-3 dietary intake were higher for adults than for children. The ω-6/ω-3 ratio was lower for the pediatric population (*p* < 0.01).

### 2.6. RBC FAs and Blood Biochemical Parameters Correlation

Different correlation profiles between RBC FAs and biochemical parameters were observed for children and adults with obesity (Figure 1 and Figure 2, respectively).

For the pediatric population inverse correlations between LA and total cholesterol and LDL cholesterol were observed, as well as an inverse correlation between EPA and triglycerides.

For adults, other correlations were observed. DHA correlated positively with AST and ALT. At the same time, oleic acid showed an inverse correlation with LDL cholesterol, and stearic acid correlated inversely with HDL cholesterol. 

### 2.7. RBC FAs and Food Groups Values Correlation

Different correlations between RBC FAs and food groups were observed for adults and children with obesity (see heatmaps in Appendix A, respectively). Children showed a positive correlation between EPA in RBC and white fish intake, and DHA in RBC with oily fish. Read meat correlated positively with cis-vaccenic acid in RBC. For adults, only positive correlations between trans fatty acids in RBC and eggs were observed.

## 3. Discussion

To our knowledge, this is the first time that a comparison of RBC membrane FA composition between adults and children with obesity has been made to determine metabolic differences, to establish dietary requirements that can contribute to design more precise nutritional strategies based on dietary fat quality to manage obesity at different age stages. The fact that the RBC membrane fatty acid composition is close to that of hepatocytes, having saturated (43% vs. 42%), monounsaturated (23.0% vs. 23.8%), polyunsaturated ω-6 (27.6 vs. 27.4%) and ω-3 (5.7% vs. 4.6%) fatty acid residues in almost similar quantities [20] is an important observation as the RBC examination avoids to run invasive investigations, especially in children.

Regarding individual nutrient intake, differences were observed between children and adults with obesity, taking into account that the nutritional recommendations for both population groups differ because metabolism requirements are different [21]. Two main reasons to measure dietary intake through a FFQ questionnaire were: to establish the eating pattern of each group and to consider dietary intake as a confounding factor in the Ancova analysis, of the study of metabolic differences, between the children and adults with obesity [22]. The elimination of the variability generated by diet in the RBC FA profile, allows us to focus on the metabolic differences between adults and children with obesity. Concerning the dietary pattern of each group, the higher levels of ω6/ω3 intake for adults with obesity was an interesting result, showing its relationship with the quality and not with the quantity of lipids intake.

Measurement of RBC FAs revealed two differentiated profiles between children and adults with obesity, where not all the differences were attributable to age, as those results have been compared with a population of adults and children with normal weight and can be due to metabolic differences.

Related with age, differences in PUFA levels between pediatric and adult, for both group with normal weight and obesity, were observed. Adults with obesity showed a proportionally, higher intake of ω6 FAs than children with obesity, contributing to a higher intake ratio of ω6/ω3, but in the RBC membrane profile, adults with obesity showed lower levels of ω6/ω3 FA ratio. Even if the children´s intake of ω6 was lower compared with adults, a higher value of ω6/ω3 FA ratio in RBC membrane was determined for children with obesity. It can be observed, that for children the contribution to ω6 levels is given, in a significantly higher manner, by linoleic and DGLA acids, whereas for ω3 levels, is given by DHA, revealing a different metabolic fate of the dietary intakes. Certainly, the greater proportion of DHA needed for heart and brain tissues [23], could be responsible for a higher distribution of this FA in children compared to the adult group, because of the growth associated with that stage of life [24]. In our opinion, levels of DHA specifically determined in cell membranes, in particular in mature RBC membranes where PUFA ω-3 were found higher than in non-selected RBC [18], should be considered as important information of the bioavailability of essential or semi-essential FA for the fundamental building up of the membrane compartment. Formation of membranes is needed for living organisms [25] and must be combined with an appropriate composition of the FA pool to avoid critical imbalances. The fatty acid–based membrane lipidomics is a diagnostic tool that provides an important piece of information in the puzzle of the metabolic pathways of health and disease [18]. 

Similar connections with mediator formation can be inferred for the ω6 FAs in RBC, as LA and DGLA levels were higher for the pediatric population compared with the adult population, indistinctly for groups with obesity and normal weight. Higher values of DGLA in children compared to adults can indicate a metabolic connection with mediators for inflammatory, immune and defense processes, since this FA is a precursor of series 1 prostaglandins, like PGE1, connected with the cAMP activity and this can occur in a higher rate in children, regardless of the intake [26].

These metabolic differences appear as important factors to evaluate the fat quality and quantity especially in diseases which involve fats, such as obesity. The adequate nutritional strategies for each population group should be personalized, for example, reinforcing the ω3 FAs intake recommendation in the pediatric population compared to the adult population, because of their higher requirements [8]. Increasing dietary intake of ω3 sources such as oily fish and seafood, especially cold-water fatty fish (sardine, tuna, salmon, mackerel) and vegetable sources such as walnuts, chia seeds, flaxseed, or even with a personalized nutraceutical plan [27] could be a roadmap.

When comparing MUFA and SFA levels in RBC, adults with obesity showed higher levels of palmitic acid and cis–vaccenic acid compared with children with obesity, while these differences were not observed among the normoweight groups (Table 1). The % of energy obtained from palmitic acid in the diet, appears in a higher proportion for children with obesity than for adults with obesity, but contrary than expected adults with obesity, reflect a higher value of RBC palmitic acid in the membrane profile. 

The higher value in the DNL index in adults with obesity compared with children with obesity, reflects a higher de novo synthesis of lipids, that can explain the higher levels of palmitic acid in adults with obesity [28]. Together with this higher DNL index, a tendency of a reduced activity of delta-9-desaturase (*p* = 0.06) for adults with obesity compared with children with obesity can be observed. This reduced activity of the enzyme is correlated to factors that have been recalled several times in the SFA pathway, as: the absence of enzymatic cofactors, the inhibition of desaturase activity and liver impairment [29]. A high–carbohydrate diet can increase rates of DNL, that has been suggested to contribute to the pathogenesis of non–alcoholic fatty liver disease (NAFLD), linked in its turn to the development of type 2 diabetes mellitus [30].

As desaturase transformation prevents SFA accumulation and toxicity triggering hepatocellular apoptosis and liver damage [31], adults with obesity, that have suffered the accumulation of SFA for years, or at least for a longer time than children, might be a plausible reason that delta-9-desaturase presents a tendency of less activity in adults than in children with obesity.

In any case, recommendations should consider these differences between adults and children, highlighting the importance of not promoting de novo synthesis in adults by lowering the intake of SFA and simple carbohydrates. Moreover, increasing the intake of PUFAs has been associated to inhibitory effects on SFA and MUFA biosynthesis [32] and could be seen as a proper recommendation.

According to biochemical parameters, even if most of them are within the optimal ranges in both groups with obesity and normal weight, it is remarkable that glucose, total cholesterol, LDL cholesterol, triglycerides and ALT/GPT showed significative lower values for children with obesity compared with adults with obesity, fact that was not observed in the group with normal weight. Additionally, differentiated correlations between biochemical parameters and RBC FAs were observed. For children, higher levels of LA inversely correlated with LDL cholesterol and total cholesterol were determined (Appendix A). This correlation has been previously reported also for both, circulating LA and RBC LA and cholesterol [33,34]. Food groups with higher content of LA, such as nuts, would be recommendable to lower LDL cholesterol levels in pediatric populations. On the other hand, adults showed an inverse correlation with oleic acid and LDL cholesterol, so higher consumption of food groups containing this FAs, such as olive oil, would be recommendable to reduce LDL cholesterol levels. SFA/MUFA ratio in adults, showed a positive correlation with LDL cholesterol, so the replacement of SFA with MUFAs in dietary intake would be advisable in order to reduce LDL cholesterol levels. For adults, DHA and total ω3 showed positive correlations with ALT and AST. Anyway, it has been reported in the literature that PUFAs can only decrease ALT and AST after long term supplementation in children [35].

As a limitation of the study, the uneven group distribution, the number of each group, should be noted even if in this type of observational studies, a perfect match is hard to achieve. At the same time, the indirect measurement of enzyme activity by the ratio between product and precursors, although very popular, could be considered as a limitation of the study and should be measured directly to emphasize and reaffirm the conclusions obtained. Another limitation of the study was the use of a population subgroup for the analysis of differences in biochemical parameters, which reduces the possibility of finding more significant differences and correlations. 

In conclusion, the present study establishes the differences of RBC FA profiles, between children and adult with obesity, demonstrating that both groups have differentiated profiles. Children with obesity present higher LA, DGLA and total ω6 values, along with lower DHA and total ω3 values, compared to adults with obesity, even after adjusting the values by their dietary intake. At the same time, children with obesity presented lower levels of palmitic acid and a lower value of the de novo lipogenesis index compared to adults with obesity. These differences must be considered to provide more specific food group recommendations based on individual FA needs, rather than giving general recommendations for a population with obesity, as a whole and regardless of age.

## 4. Materials and Methods

### 4.1. Subjects and Study Design

An observational, case-control, and retrospective study was conducted on 83 children with obesity (26 boys and 57 girls) between 6 to 16 years old and a group of 61 adults with obesity (19 males and 42 females) between 19 to 68 years old, recruited from pediatric endocrinology and the endocrinology department at the Hospital Universitario Cruces (Barakaldo, Spain). Control subjects, consisting of 113 normoweight children and 30 normoweight adults, were also recruited from the same centers as patients with obesity. Children were classified according to body mass index (BMI), using age and sex-specific pediatric z-scores from Orbegozo tables [19]. The BMI was taken as a reference to define the different categories, defining normal weight when the standard deviation (SD) of BMI was −1 < SD ≤ +1, overweight when +1 < SD ≤ +2, and obesity when SD > +2. For adults, BMI > 30 Kg/m^2^ was taken as a reference to classify obesity and 18.5 < BMI < 25 Kg/m^2^ for the normoweight group.

Subjects were excluded if they presented any kind of acute or chronic diseases, were taking medications, had any presence of metabolic syndrome symptoms, or obesity-associated with any type of pathology. A physical examination was performed by an endocrinologist. 

The study protocol was approved by the Euskadi Clinical Research Ethics Committee (permission number PI2016181) and carried out according to the Declaration of Helsinki Good Clinical Practice guidelines. Subjects under study were included after acceptance (by the parents in the case of the pediatric population) to participate in the study and signing of informed consent. In the case of children between 12–16 years of age the informed consent was also signed by themselves according to the Euskadi Ethical Committee and sample biobank laws (Organic Law 3/2018, of December 5, on Protection of Personal Data and guarantee of digital rights; Law 14/2007 on Biomedical Research and RD 1716/2011 of Biobanks).

### 4.2. Anthropometric Measures

Bodyweight (kg) and height (cm) were measured by standardized methods [36]. Body mass index (BMI) was calculated as weight (kg) divided by the square of the height (m^2^). Anthropometric parameters, as well as blood sampling, were all conducted by pediatricians and doctors during the participant’s visit to the Hospital Universitario Cruces/IIS Biocruces Bizkaia.

### 4.3. Nutrient Intakes

During the participant’s visit with the endocrinologist, the doctor interviewed the participants and collected personal data, including family medical history and information on the history of medication usage. Estimations of food consumption, including dietary diversity and variety, were measured using a quantitative food frequency questionnaire (FFQ) on-line completed by the parents of the children, except in those cases of adolescents, which were encouraged to complete it themselves, or by each adult volunteer. For our study, an adapted FFQ was used, which was previously validated with the portion sizes and food groups for the Spanish juvenile population and adults [37,38]. These questionnaires were then analyzed using the DIAL^®^ software (UCM & Alce Ingeniería S.A., Madrid, Spain) (V 3.4.0.10) to translate the intake of specific foods into their corresponding energy and nutrient values. 

### 4.4. Red Blood Cell (RBC) Membrane Fatty Acid Analysis

The fatty acid composition of mature RBC membrane phospholipids was obtained from blood samples (approximately 2 mL) collected in vacutainer tubes containing ethylenediaminetetraacetic acid (EDTA). Samples were shipped to the Lipidomic Laboratory approved for the method by the UNI CEI EN ISO/EIC 17025:2018 (#1836L belonging to the company Lipinutragen, Bologna, Italy) and upon arrival underwent the certified procedure MEM_LIP_1 according to the quality control guidelines. At first, the absence of hemolysis was checked upon arrival. From the blood, the protocol consists of the selection of mature RBCs by a robotic platform, as reported previously [17,22,39,40], followed by lipid extraction and lipid transesterification to fatty acid methyl esters (FAMEs). Briefly, the whole blood in EDTA was centrifuged (4000 rpm for 5 min at 4 °C), and the mature cell fraction was isolated by the robotic platform, based on the higher density of the aged cells [41], and checked by the use of cell counter (Scepter 2.0, EMD Millipore, Darmstadt, Germany). The automation included cell lysis, isolation of the membrane pellets, phospholipid extraction from pellets using the Bligh and Dyer method [42], transesterification to FAMEs by treatment with a potassium hydroxide (KOH)/methyl alcohol (MeOH) solution (0.5 mol/L) for 10 min at room temperature, and extraction using hexane (2 mL). The final FAME mixtures were analyzed using capillary column gas chromatography (GC). GC analysis was run on the Agilent 6850 Network GC System, equipped with a fused silica capillary column Agilent DB23 (60 m × 0.25 mm × 0.25 μm) and a flame ionization detector. Optimal separation of all fatty acids and their geometrical and positional isomers was achieved. Identification and quantification of each fatty acid were made by calibrated procedures that are part of the MEM_LIP_1 method. Commercially available standards and a library of trans isomers of MUFAs and polyunsaturated fatty acids (PUFA) were used as standards. The amount of each FA was calculated as a quantitative percentage of the total FA content (relative quantitative %), as described in Section 4.5, being more than 97% of the GC peaks recognized with appropriate standards. The use of mass spectrometry is only at the level of comparing the LIBRARY of fatty acid standard references mass data with the GC peaks and masses obtained from the samples.

### 4.5. Red Blood Cell Membrane Fatty Acid Cluster

12 FAs were selected as a representative cluster of the dominant glycerophospholipids present in the RBC membrane, as well as three FA families (SFA, MUFA and PUFA): for SFAs, palmitic acid (C16:0) and stearic acid (C18:0); for MUFAs, palmitoleic acid (C16:1;9c), oleic acid (C18:1; 9c), cis-vaccenic acid (C18:1; 11c); for ω-3 PUFAs, eicosapentaenoic acid (EPA) (C20:5), docosahexaenoic acid (DHA) (C22:6); for ω-6 PUFAs, linoleic acid (LA) (C18:2), dihomo–gamma–linolenic acid (DGLA) (C20:3) and arachidonic acid (AA) (C20:4); for geometrical trans fatty acids (TFA): elaidic acid (C18:1 9t) and mono–trans arachidonic acid isomers (monotrans-C20:4; ω-6 recognized by standard references as previously described by Ferreri et al. [43]. Considering these fatty acids, different indexes previously reported in the literature [44] were calculated: (%SFA/%MUFA) index related with membrane rigidity; Omega-3 index (DHA + EPA); Inflammatory risk index (%ω-6)/(%ω-3); PUFA balance [(%EPA + %DHA)/total PUFA × 100]; Free radical stress index (sum of trans-18:1 + Σ monotrans 20:4 isomers); Unsaturation Index (UI) [(%MUFA) + (%LA/2) + (%DGLA/3) + (%AA/4) + (% EPA/5) + (%DHA/6)]; Peroxidation Index (PI) [(%MUFA/0.025) + (%LA) + (%DGLA/2) + (%AA/4) + (% EPA/6) + (%DHA/8)]; De Novo Lipogenesis index (DNL) [(%Palmitic acid)/(%LA)] [45].

Additionally, the enzymatic indexes of elongase and desaturase enzymes, the two classes of enzymes of the MUFA and PUFA biosynthetic pathways, were inferred by calculating the product/precursor ratio of the FAs involved in these reactions.

### 4.6. Biochemical Parameters

Blood biochemical parameters were measured with standard laboratory assays after collecting venous blood samples performed in the morning in fasting state from a subgroup of the studied population (69 children vs. 44 adults with obesity and 34 pediatric vs. 30 adults with normal weight). Plasma concentrations of glucose, serum concentrations of total cholesterol (TC), high–density lipoprotein cholesterol (HDL-C), low–density lipoprotein cholesterol (LDL-C), triglycerides (TG), aspartate aminotransferase (AST), alanine aminotransferase (ALT), uric acid, and bilirubin were measured.

### 4.7. Statistical Analysis

Differences between groups for the nutrient intake and biochemical values were determined by using the Mann–Whitney U test for data that were not normally distributed and the t–student test for normally distributed variables. Normal data distribution was assessed by Shapiro–Wilk’s test and Kolmogorov–Smirnov test.

An ANCOVA was run to determine the differences between RBC membrane fatty acids from children and adults with obesity, and also for normoweight children and normoweight adults, after controlling for variables selected as potential confounders, such as gender, BMI, and dietary intake [45]. Post hoc analysis was performed with a Bonferroni adjustment. First, a Principal Component Analysis (PCA) was run on 15 dietary nutrient intake variables (individual FAs, families (SFA, MUFA, and PUFA), total lipids, carbohydrates, and proteins) obtained with the DIAL software (v3.4.0.10, Department of Nutrition (UCM) & Alce Ingeniería, S.L., Madrid, Spain) after transforming the information about food items from FFQ questionnaires into micro and macronutrient values, to reduce and simplify the dimension of these variables and use the generated factors as diet covariates [22]. The Kaiser–Meyer–Olkin (KMO) and Bartlett’s test of sphericity were used to verify the sampling adequacy for the analysis. PCA revealed four components that had eigenvalues greater than one and which explained 82.55% of the total variance. These components were included in the ANCOVA analysis as diet covariates. The level of significance was set at *p* < 0.05.

Correlations between the RBC membrane FA profile and dietary intake and FA profile and biochemical values were performed using Spearman’s rank-order correlation coefficients (ρ) and *p*-values were adjusted with the false discovery rate method for multiple comparisons. Correlation plots were visualized using the R heatmap.2() function. All other statistical analyses were performed using SPSS (IBM Corp. v24.0, Armonk, NY, USA).

## Figures and Tables

**Figure 1 metabolites-11-00043-f001:**
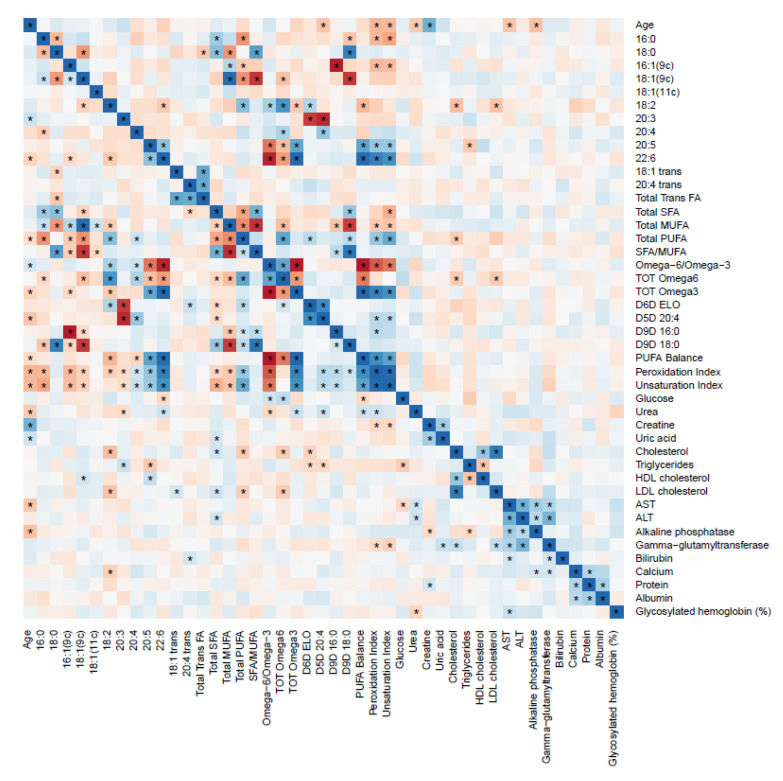
Heat map of the correlations between red blood cell membrane fatty acids and biochemical parameters for children with obesity. The color represents the Spearman correlation coefficient (ρ) (blue = positive; red = negative). * Represent significant correlations (*p* < 0.05) between variables. The pair variables that have an * above and below the diagonal line are significantly correlated after correction for multiple comparisons (*q* < 0.05).

**Figure 2 metabolites-11-00043-f002:**
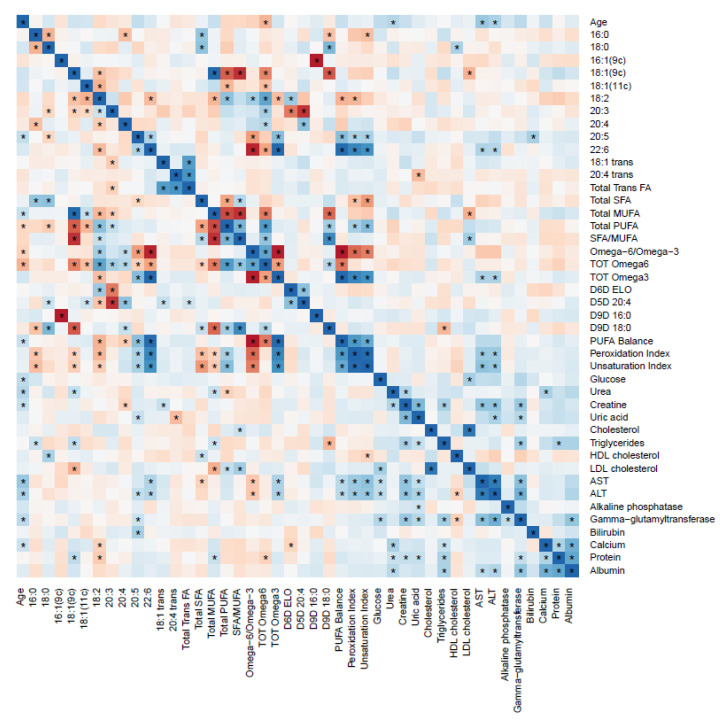
Heat map of the correlations between red blood cell membrane fatty acids and biochemical parameters for adults with obesity. The color represents the Spearman correlation coefficient (ρ) (blue = positive; red = negative). * Represent significant correlations (*p* < 0.05) between variables. The pair variables that have an * above and below the diagonal line are significantly correlated after correction for multiple comparisons (*q* < 0.05).

**Table 1 metabolites-11-00043-t001:** Red blood cell **(**RBC) membrane fatty acid profile.

	Group with Obesity					Group with Normal Weight				
	Pediatric *n* = 83		Adult *n* = 61		Ancova	Pediatric *n* = 113		Adult *n* = 30		Ancova
Fatty Acid (%)	Mean	SE	Mean	SE	*p* *	Mean	SE	Mean	SE	*p* *
Palmitic acid (C16:0)	22.31	0.16	23.22	0.22	0.01	22.51	0.10	22.72	0.23	0.43
Stearic acid (C18:0)	18.22	0.18	17.54	0.24	0.06	17.68	0.10	17.66	0.23	0.94
TOTAL SFA	40.55	0.16	40.78	0.21	0.48	40.21	0.10	40.22	0.23	0.97
Palmitoleic acid (C16:1)	0.46	0.02	0.37	0.03	0.08	0.41	0.02	0.48	0.03	0.08
Oleic acid (9c C18:1)	16.55	0.20	17.08	0.27	0.20	17.46	0.12	17.79	0.27	0.29
cis–Vaccenic acid (11c C18:1)	1.17	0.04	1.34	0.06	0.05	1.22	0.02	1.32	0.05	0.07
TOTAL MUFA	18.19	0.21	18.78	0.29	0.18	19.10	0.13	19.57	0.28	0.17
Linoleic acid (C18:2)	14.00	0.27	12.39	0.37	0.01	14.22	0.12	13.12	0.27	<0.01
DGLA (C20:3)	2.35	0.07	2.07	0.09	0.04	2.05	0.04	1.81	0.08	0.02
ARA (C20:4)	19.73	0.21	19.39	0.29	0.44	18.75	0.14	18.32	0.32	0.26
TOTAL ω6	36.12	0.29	33.84	0.39	<0.01	35.02	0.16	33.27	0.35	<0.01
EPA (C20:5)	0.49	0.04	0.63	0.06	0.10	0.59	0.02	0.65	0.05	0.31
DHA (C22:6)	4.52	0.19	5.84	0.26	<0.01	4.93	0.10	5.83	0.23	<0.01
TOTAL ω3	5.01	0.21	6.47	0.29	<0.01	5.52	0.11	6.48	0.26	<0.01
TOTAL PUFA	41.12	0.24	40.31	0.32	0.10	40.54	0.15	39.88	0.33	0.09
Trans C18:1	0.08	0.01	0.07	0.01	0.65	0.08	0.01	0.10	0.02	0.34
Trans C20:4	0.07	0.01	0.08	0.01	0.59	0.07	0.01	0.10	0.02	0.14
TOTAL TRANS	0.15	0.01	0.15	0.02	0.98	0.16	0.01	0.14	0.02	0.46
**Indexes**										
ω6/ω3	7.55	0.27	5.51	0.37	<0.01	6.66	0.15	5.11	0.34	<0.01
SFA/MUFA	2.24	0.03	2.19	0.04	0.37	2.12	0.02	2.07	0.04	0.32
Omega–3 Index	5.01	0.21	6.47	0.29	<0.01	5.52	0.11	6.48	0.26	<0.01
∆6D+ELO 20:3/18:2 ^a^	0.17	0.006	0.17	0.008	0.96	0.14	0.003	0.14	0.006	–
∆5D 20:4/20:3	8.60	0.31	9.51	0.41	0.15	9.45	0.21	10.55	0.47	0.05
∆9D 16:1/16:0	0.02	0.001	0.016	0.001	0.06	0.018	0.001	0.02	0.001	–
∆9D 18:1/18:0	0.91	0.02	0.98	0.02	0.07	0.99	0.01	1.00	0.01	0.67
DNL Index 16:0/18:2	1.62	0.03	1.85	0.05	<0.01	1.59	0.02	1.72	0.03	<0.01
PUFA BALANCE	12.13	0.52	16.09	0.70	<0.01	13.61	0.27	16.21	0.61	<0.01
Peroxidation Index	136.77	1.42	145.07	1.91	0.01	136.81	0.82	141.31	1.89	0.04
Unsaturation index	161.53	0.92	165.51	1.24	0.04	161.25	0.61	163.57	1.37	0.15

Data are presented as mean ± standard error (SE). * Adjusted for age, sex, and dietary components, extracted from the principal component analysis of dietary nutrient intake (individual FAs, families (SFA, MUFA, and PUFA), total lipids, carbohydrates, and proteins). Post hoc tests were conducted with a Bonferroni adjustment. ^a^ Levene’s test of homogeneity of variance was not met.

**Table 2 metabolites-11-00043-t002:** Biochemical values measured in plasma in a fraction of the observed groups.

	Group with Obesity			Group with Normal Weight		
	Pediatric *n* = 69	Adult *n* = 44	*p* *	Pediatric *n* = 34	Adult *n* = 30	*p* *
	Med (Q1–Q3)	Med (Q1–Q3)		Med (Q1–Q3)	Med (Q1–Q3)	
Glucose (mg/dL)	85 (79–89.25)	97 (90.5–107.5)	<0.01	84 (81–89)	85 (79.75–92)	0.38
Uric Acid (mg/dL)	4.95 (4.375–5.7)	5.6 (4.9–6.95)	<0.01	3.95 (3.37–4.62)	4.75 (3.8–5.22)	0.03
Total Cholesterol (mg/dL)	150 (132.7–172)	180 (158–211)	<0.01	165 (148.5–186.7)	176.5 (141.7–206.2)	0.46
Triglycerides (mg/dL)	76 (55.5–108.7)	123 (89.5–180.5)	<0.01	65.5 (46–86)	68 (58.75–84.75)	0.48
HDL cholesterol (mg/dL)	44.6 (40.0–54.25)	47 (41.75–56)	0.32	55 (48.5–64.5)	59 (48.5–71)	0.48
LDL cholesterol (mg/dL)	88.4 (71.25–98)	118 (95–141)	<0.01	95 (77–110)	102 (72.5–121)	0.35
AST/GOT (U/L)	22 (19–26.25)	20 (16.5–26.5)	0.12	26 (22–27)	19 (16.75–23.5)	<0.01
ALT/GPT (U/L)	18.5 (15–23.25)	23 (15–35.5)	0.03	16 (13.75–18)	17 (12.75–21.25)	0.59
Bilirubin (mg/dL)	0.4 (0.3–0.6)	0.4 (0.2–0.5)	0.5	0.6 (0.4–1)	0.4 (0–0.625)	0.01

Data expressed as medians and quartile 1 and quartile 3 (Med (Q1–Q3)). * Not normally distributed variables. A Mann-Whitney U test was carried out.

**Table 3 metabolites-11-00043-t003:** Macronutrients and individual fatty acids (FA) intake expressed as % energy (%*E*) in pediatric and adult groups with obesity.

	Pediatric Group with Obesity, *n* = 83		Adult Group with Obesity, *n* = 61		Mann-Whitney U Test
	Mean	SD	Mean	SD	*p*
**Macronutrient**					
Calories (Kcal/day)	2044.1	564.3	2480.1	794.2	<0.01
Proteins (%E)	16.5	2.1	16.3	3.2	0.54
Carbohydrates (%E)	46.7	5.3	36.3	6.6	<0.01 *
Simple sugars (%E)	21.7	4.9	19.1	6.4	<0.01
Lipids (%E)	33.6	6.3	42.6	6.3	<0.01 *
**Individual Fatty Acids**					
C14:0	1.0	0.5	0.8	0.3	0.08
C16:0	6.1	1.3	5.6	1.0	0.02 *
C18:0	2.3	0.6	2.3	0.5	0.93
Total SFA	10.8	2.9	11.2	2.2	0.09
C16:1	0.5	0.1	0.5	0.1	0.86
C18:1	14.1	3.5	18.9	4.1	<0.01 *
Total MUFA	15.0	3.6	19.9	4.2	<0.01 *
C18:2	4.1	1.7	6.9	2.5	<0.01
C20:4	0.04	0.01	0.08	0.04	<0.01
Total ω6	4.1	1.7	7.0	2.5	<0.01
C18:3	0.54	0.13	0.79	0.34	<0.01
C20:5 (EPA)	0.07	0.06	0.07	0.05	0.24
C22:5 (DPA)	0.02	0.01	0.02	0.01	0.48
22:6 (DHA)	0.15	0.09	0.14	0.08	0.62
Total ω3	0.79	0.22	1.03	0.39	<0.01
Total PUFA	5.1	1.8	8.2	2.7	<0.01
ω6/ω3	5.5	2.2	7.2	2.8	<0.01

Data presented as mean and standard deviation (SD). Not normally distributed variables. A Mann–Whitney U test was carried out. * Normally distributed variables, an independent-samples t-test was performed.

## Data Availability

Authors should ensure that data shared are in accordance with consent provided by participants on the use of confidential data. The data presented in this study are available on request from the corresponding author (S.A.). Raw data were generated at AZTI, Biocruces Bizkaia Health Research Institute and Lipidomic Laboratory maintaining samples anonymized.

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
