# Peer review of "Molecular Differences Based on Erythrocyte Fatty Acid Profile to Personalize Dietary Strategies between Adults and Children with Obesity"

_metabolites, 2021, doi:10.3390/metabo11010043_

Round 1

Reviewer 1 Report

The study “Molecular differences based on erythrocyte fatty acid profile to personalize dietary strategies between adults and children with obesity“ submitted from Sara Arranz provides a thorough analysis of the overall lipid profile of the membranes of mature erythrocytes dependent on the individual age and nutritional state with the aim to find causal relationships between function and the fatty acid composition as well as differentiated nutrition strategies for the affected obese people.

I am not aware of a similar systematic study addressing the relationship between the plasma membrane fatty acid composition and age/obese state. Thus albeit the results obtained are not very surprising, e.g. that glucose and LDL cholesterol levels are lower in obese children compared to obese adults, the study is of potential interest for the readers of Metabolites, in particular, for those engaged in basic and clinical research of metabolic diseases. Nevertheless, the following points should be addressed by the authors:

  1. Please present the patient characteristics in greater detail, perhaps by adding a supplementary table
  2. Are data available for probands with overweight, i.e. BMI between 25 and 30
  3. Are data available for the metabolic state of the probands, i.e. glucose, insulin, insulin sensitivity (oral glucose tolerance)
  4. In this case, are there any correlations detectable between the metabolic state and the RBC membrane fatty acid composition, which are dependent on or independent of age and/or weight
  5. I would like to see some of the heat maps in the main text rather than as supplementary figure (e.g. S1 and S2).
  6. I do not recognize the experimental procedure and the results for determination of the de novo lipogenesis index. Please add corresponding passages in the Methods and Results
            1.  

Author Response

1. Please present the patient characteristics in greater detail, perhaps by adding a supplementary table

It has been added in the supplementary material, Table S1, as for patients characteristics, only age, BMI (which is only used in adults as children use the Z-score) and gender are available.

2. Are data available for probands with overweight, i.e. BMI between 25 and 30

Unfortunately, not in this study.

3. Are data available for the metabolic state of the probands, i.e. glucose, insulin, insulin sensitivity (oral glucose tolerance)

We have measured glucose levels as it is presented in Table 2 that shows blood biochemical parameters, but we have not measured insulin or insulin sensitivity. We agree that it would be interesting to include it in future studies.  

4. In this case, are there any correlations detectable between the metabolic state and the RBC membrane fatty acid composition, which are dependent on or independent of age and/or weight

Correlations between blood biochemical parameters and RBC membrane fatty acid composition were represented in previous Figures S1 and S2 for both children and adults respectively. New heatmaps of correlations between biochemical parameters and RBC fatty acids including age, for each group (children and adults) have been now included in the manuscript as Figures 1 and 2.

 5. I would like to see some of the heat maps in the main text rather than as supplementary figure (e.g. S1 and S2).

Figure S1 and Figure S2 showing correlations between RBC membrane fatty acid composition and blood biochemical parameters have been included in the Results section as Figure 1 and Figure 2, respectively.

6. I do not recognize the experimental procedure and the results for determination of the de novo lipogenesis index. Please add corresponding passages in the Methods and Results

It is described in the Methods section, line 161, with a referenced study (reference 29) to check his suitability. It is determined by the ratio between palmitic acid and linoleic acid.

Reviewer 2 Report

In this study, Jauregibeitia et al. aimed to investigate the lipid profile of RBC membranes. They undertook mass spectrometry analyses of RBC isolated from lean and obese children, as well as adults. They used their data to suggest some nutritional guidelines for children and adults to protect against obesity.

Overall, it unclear to me what the significance of the work is and how it adds to potential treatments / nutritional guidelines for obesity.

In particular:

  1. Why do the authors choose to focus on RBCs, in particular those that are old and near the end of their life? How are RBCs more informative for obesity research, than say the serum which actually represents the global metabolic profile? What are the advantages of RBCs over serum?

  1. The main mass spectrometry methodology is only vaguely described. From what I can gather of the methodology, all lipids in RBCs (and not only those at the plasma membrane) will be broken down with this methodology. Furthermore, free fatty acids will also be analyzed. So, can the authors explain how they specifically enrich for the membranes? Why have the authors chosen to break down all complex lipid species rather than measure complex lipids?

  1. How can the authors rule out that the differences they see in the lipid profiles are not due to different lipid processing pathways present in adults versus children? Or fatty acid processing pathways impacted by obesity? Or by improper ROS balance that lead to oxidation of fatty acids / lipids?

  1. If the major aim of this study is to suggest nutritional interventions for obesity, why are the authors comparing results in obese children versus adults? Surely, there will be significant differences simply due to age! The data should really be compared between the obese versus healthy states in adults and separately in children for any meaningful conclusions.

  1. With regards to the nutritional survey – it simply shows that the dietary intake is different between adults and children (regardless of obesity). What is the biological or physiological significance of this? How does this help understand obesity in any way?

  1. Why did the authors choose to study old RBCs rather than young ones? As the RBCs and their membranes would have formed at least 3 months prior to analyses, why do the authors think they represent the metabolic state of the body?

  1. Is there any evidence that changing the intake of specific fatty acids improves weight loss or obesity status?

Author Response

1. Why do the authors choose to focus on RBCs, in particular those that are old and near the end of their life? How are RBCs more informative for obesity research, than say the serum which actually represents the global metabolic profile? What are the advantages of RBCs over serum?

We explained in the Introduction this choice, citing 4 references that related to the work done in this field.

Here we give other details of the approach which is based on two main considerations: a) the role of fatty acid residues in membrane phospholipids which has quite a different meaning compared to fatty acids of circulating lipids; the former derive from a combination of nutritional AND metabolic effects due to “stabilized dietary habits” and represent also systemic effects like membrane properties and signaling cascades, whereas the latter ones are more influenced by the short-term intakes of dietary fats and clearly do not have information of membrane formation and activity.

The red blood cells have been described several times for the role of the fatty acids of their membranes and, more recently, the methodology of separation and analysis of mature RBC membrane fatty acids has reached a high level of precision (ISO/EIC 17025). The main reasons to use mature RBC are connected with precision of the sampling and of the results: i) the first reason is that the selection of mature RBC can be done by centrifugation and by checking their diameter (as performed in the ISO17025 approved methodology); ii) RBCs have a quite long life  (4 months average lifetime) and they exchange phospholipids with tissues and lipoproteins, modifying their composition during their life, therefore they represent very well the conditions of other tissues; iii) in particular they show higher levels of PUFA in their maturity (Diagnostics 2017) therefore  this information is very important for the assessment of PUFA levels in health and disease, especially in obesity which can include inflammatory conditions; iv) more in general, when measurements of biomolecules are involved, it is necessary to make strategic decision on the biological significance of the measured parameters and clinical use.

Fatty acids in cell membranes are the most interesting to evaluate since by dietary intervention the fatty acid plasma levels are certainly changed, but the “systemic effects” are obtained if dietary lipids reach cell membranes.

As far as the “near to the end” question of this referee, RBCs maintain their properties until the last moment of their life, and there is no indication of loss of properties “near to the end”. Actually, it is an intriguing and still not answered question why RBCs die!

2. The main mass spectrometry methodology is only vaguely described. From what I can gather of the methodology, all lipids in RBCs (and not only those at the plasma membrane) will be broken down with this methodology. Furthermore, free fatty acids will also be analyzed. So, can the authors explain how they specifically enrich for the membranes? Why have the authors chosen to break down all complex lipid species rather than measure complex lipids?

In the Experimental section we detailed the steps of this analysis and we referred to a certified methodology (ISO17025) to separate mature RBC membranes and obtain phospholipids by lipid extraction. It is worth mentioning that our methodology clearly isolates as fatty acid methyl esters (FAME) contained in all glycerophospholipids of mature RBC membranes, where fatty acids are esterified with the glycerol moiety are examined (phosphatidyl choline, phosphatidyl ethanolamine, phosphatidyl serine, plasmalogens).

This eliminates all other possible interferences by free fatty acids and other lipids contained in the blood. The only other lipid present is cholesterol, but this is clearly separated and identified from fatty acids.

Secondly, the use of mass spectrometry is only at the level of comparing the LIBRARY of fatty acid standard references mass data with the GC peaks and masses obtained from the samples. We added this detail in the end of par 2.4 (rows 144-146).

3. How can the authors rule out that the differences they see in the lipid profiles are not due to different lipid processing pathways present in adults versus children? Or fatty acid processing pathways impacted by obesity? Or by improper ROS balance that lead to oxidation of fatty acids / lipids?

The most important comparisons of the work is not only between obese adults and children, but also the differences between normal weight adults and children, as shown in Table 1. In fact, the relevance of the palmitic acid level, and consequently of the decreased delta-9 desaturase transformation and unsaturation index, is emerging from this comparison which, in our knowledge, has been performed for the first time.

The important fact that in children with obesity the desaturase enzyme is working less to create palmitoleic acid is an important indication, due to the role of adipokine of this fatty acid.

4. If the major aim of this study is to suggest nutritional interventions for obesity, why are the authors comparing results in obese children versus adults? Surely, there will be significant differences simply due to age! The data should really be compared between the obese versus healthy states in adults and separately in children for any meaningful conclusions.

The influence of age has been taken into account by the comparison of healthy adults and children.

5. With regards to the nutritional survey – it simply shows that the dietary intake is different between adults and children (regardless of obesity). What is the biological or physiological significance of this? How does this help understand obesity in any way?

The nutritional survey is always necessary when populations are compared, in order to exclude or include the role of specific food intake. On the other hand, we use this dietary information as a covariate for the Ancova analysis, as it is explained in the Methods section, rows 177-185.

6. Why did the authors choose to study old RBCs rather than young ones? As the RBCs and their membranes would have formed at least 3 months prior to analyses, why do the authors think they represent the metabolic state of the body?

RBC created a real phospholipid molecular flux with the other tissues and this process occurs aling the RBC lifetime, therefore older RBCs are more “experienced” for this. The phospholipid exchange is a very important, well assessed process occurring in the body (Reed, C.F. Phospholipid exchange between plasma and erythrocytes in man and the dog. J. Clin. Investig. 1968, 47, 749–760, Dushianthan, A.; Cusack, R.; Koster, G.; Grocott, M.P.W.; Postle, A.D. Insight into erythrocyte phospholipid molecular flux in healthy humans and in patients with acute respiratory distress syndrome. PLoS ONE 2019, 14, e02215959)

7. Is there any evidence that changing the intake of specific fatty acids improves weight loss or obesity status?

We believe that the personalization of the dietary fatty acids in overweight/obese people has not yet addressed using our approach of mature RBC membranes, therefore we suggest the implementation of nutritional intervention studies for this scope.

Reviewer 3 Report

This is an interesting paper, especially from practical point of view, indicating lipid profile differences in mature red blood cells membranes between children and adult with obesity. However, there are some major and minor issues that need to be addressed:

Major issues.

Both children group and adult group with obesity are not homogenous. The study was performed on: a) children with obesity between 6 to 16 years old; b) adults with obesity between 19 to 68 years. For instance the authors showed that children group with obesity exhibit lower levels of palmitic acid and cis-vaccenic acid compared to the adult group with obesity. Considering that children and adults groups are highly heterogeneous, one can suppose that similar differences could be observed between very young children (6-7 years old) and older children (15-16 years old) or between young (18 -25 years old)  and older adults (50-68). To confirm (or not) this supposition, the authors should check correlation between palmitic acid (and cis-vaccenic acid or other parameters examined) and age of children and adults.

Minor issues.

1.In general blood cholesterol and triacylglycerols concentration are usually higher in older than in young people. I wonder why no differences between these  groups (with normal weight ) were observed by the authors. Could you explain this problem?

Author Response

This is an interesting paper, especially from practical point of view, indicating lipid profile differences in mature red blood cells membranes between children and adult with obesity. However, there are some major and minor issues that need to be addressed:

Major issues:

Both children group and adult group with obesity are not homogenous. The study was performed on: a) children with obesity between 6 to 16 years old; b) adults with obesity between 19 to 68 years. For instance, the authors showed that children group with obesity exhibit lower levels of palmitic acid and cis-vaccenic acid compared to the adult group with obesity. Considering that children and adults groups are highly heterogeneous, one can suppose that similar differences could be observed between very young children (6-7 years old) and older children (15-16 years old) or between young (18 -25 years old) and older adults (50-68). To confirm (or not) this supposition, the authors should check correlation between palmitic acid (and cis-vaccenic acid or other parameters examined) and age of children and adults.

A Spearman rank-order correlation was performed, due to the non-normality of the data of most of the variables, separately for children and adults with obesity, between the age and PLM.

For children with obesity, a positive correlation between stearic acid and DGLA was observed (p=0.016 and p=0.03) and a negative correlation for DHA (p=0.013) with age. After multiple comparison correction, by false discovery rate method, none of the observed differences were statistically significant anymore.

For adults with obesity, same methodology was followed. With the spearman correlation, only EPA showed a significative positive correlation with age (p= 0.016). After multiple comparison correction, by false discovery rate method,

Minor issues:

1.In general blood cholesterol and triacylglycerols concentration are usually higher in older than in young people. I wonder why no differences between these groups (with normal weight) were observed by the authors. Could you explain this problem?

Authors agree with revisor’s comment that increased values of tryglicerides and total cholesterol are usually observed in adults, more than in children, however dyslipidemia is a multifactorial factor that arises from changes in lifestyle, including inappropriate diet and physical inactivity, factors also strongly associated with obesity. Moreover, the authors recognize the fact of having uneven group distribution and low number of participants as a limitation as we have described in the Discussion part (rows 379-380). If we look closely at the ranges of the quartiles, for blood biochemical parameters, a wide variation can be observed with a non-normal distribution, suggesting that there is still a need for increasing sample size. Nevertheless, we consider that these results would be interesting to include in our manuscript.

Reviewer 4 Report

Dear Authors,

I found your study to be interesting, but I do have some notes that I believe that could improve your manuscript.

  • Please check thoroughly the manuscript for typos, grammar and spelling (e.g. Row 23: “61with”, Row 38 “obesity(1)”, Row 56: “metabolism(6)”, Row 57: “increasing” – to increase?, Row 58: “that,”, Row 61: “from” – regarding?, Row 81: “adult” – adults?, Row 90: “Barakaldo.Spain”, Row 103: “under” – in the?, Row 103: “acceptance” – consent?, Row 114: “2.3. Nutrient Intakes” – Nutritional intake?, Row 226: “Blood biochemistry parameters” – Blood biochemical parameters?, Row 310: “intakes” -intake?, Row 318: “unbalances” – imbalances?, Row 324: “for” – of?, Row 326: “and this” – which?, Row 327: “intake” – dietary intake?, Row 339: remove “proportion”,  etc), especially the way references are inserted (before, after “.”, with or without proper spacing, etc);
  • Rows 20-30: please rephrase “These differences should be considered to provide specific nutritional recommendations for different age groups”;
  • Row 44: “Low-fat diets, to reduce calory intake” – rephrase? “aimed at reducing caloric intake", perhaps? However, is caloric restriction the main goal of low-fat diets, as most cardiovascular disease guidelines recommend low-fat diets?;
  • Rows 52-55: “Other dietary plans have been proposed including low-carbohydrate diets, high-protein diets, very low-caloric diets with meal replacements, Mediterranean diet, and diets with intermittent energy restrictions, evidencing that a successful diet must be healthy,balanced and without nutritional deficiencies” – please rephrase for more clarity;
  • Rows 83-84: “giving relevance to the functional roles of the different fatty acid residues in lipids” – rephrase?
  • Regarding abbreviations, once they are presented in the text, please do not alternate abbreviation-full form; e.g. Row 205: “Linoleic acid” and “LA” in Row 214, or “Unsaturated Index” in Row 216, when the abbreviation should be used. Similarly, in sections 3.5 and 3.6 is it “FAs” or “FA-s”? Please double check and use the same abbreviation. Row 345: “D9D” – abbreviation not previously presented and in Row 348: “desaturase” - if you introduce the abbreviation delta-9-desaturase - D9D in the Materials and Methods section, you should further use it. If not, please replace D9D in line 345 and 354.
  • Table 1: p=0.06 for “Stearic acid” – I think it would be interesting the state (and maybe discuss/look for other literature reports) that a borderline significance level was observed for stearic acid, values being higher for children (in opposition the palmitic acid) (?)
  • Could you motivate why you have chosen to assess the routine blood biochemical parameters only for a subsample of your study population? I find this to be a serious liability of your study.
  • Table 2: I believe you should also analyze the obese vs normal weight for both children and adults. You should firstly emphasize the differences between normal and obese for each age group in your study and then obese-obese and normal-normal. For example, TC and HDL are higher in non-obese children, but LDL is lower in obese. Are the differences significant? Maybe these data should be taken into consideration when you discuss the results of your study.
  • A small note – I appreciate the thorough statistical analysis applied.
  • Rows 243-244: Again, here it might be useful to compare the two children groups (obese and non-obese) and the two adults’ groups. Comparing the intake (g) of children and adults, when their caloric necessities are different seems pointless and the p-values presented in the S1 table are irrelevant, in my opinion, and you could directly present/discuss Table 3.
  • Row 255: “the pediatric population showed higher intake of stearic acid” – Was this previously reported? Are other literature sources that state the same? Is there an explanation?
  • Table 3 – formatting issue: in the table head row the font for “Mann-Whitney U test” seems to be different from the one of the previous two column heads.
  • Row 297: “higher levels of w6/w3 ratio in RBCs for adults” - please double check and clarity, as in Table 1, the higher values are presented for obese children.
  • Rows 300-301: “have been compared with a population of adults and children with normal weight” – As stated above, firstly, you should check if RBC profiles differ in normal vs obese groups, not just obese vs obese and normal vs normal, or, please indicate some previous studies that have already reported and established such differences. Or… Please explain why you chose not to make the suggested comparisons.

Author Response

- Please check thoroughly the manuscript for typos, grammar and spelling (e.g. Row 23: “61with”, Row 38 “obesity(1)”, Row 56: “metabolism(6)”, Row 57: “increasing” – to increase?, Row 58: “that,”, Row 61: “from” – regarding?, Row 81: “adult” – adults?, Row 90: “Barakaldo.Spain”, Row 103: “under” – in the?, Row 103: “acceptance” – consent?, Row 114: “2.3. Nutrient Intakes” – Nutritional intake?, Row 226: “Blood biochemistry parameters” – Blood biochemical parameters?, Row 310: “intakes” -intake?, Row 318: “unbalances” – imbalances?, Row 324: “for” – of?, Row 326: “and this” – which?, Row 327: “intake” – dietary intake?, Row 339: remove “proportion”,  etc), especially the way references are inserted (before, after “.”, with or without proper spacing, etc);

Authors have reviewed and changed all corrections in the manuscript.

- Rows 20-30: please rephrase “These differences should be considered to provide specific nutritional recommendations for different age groups”;

This sentence has been changed.

- Row 44: “Low-fat diets, to reduce calory intake” – rephrase? “aimed at reducing caloric intake", perhaps? However, is caloric restriction the main goal of low-fat diets, as most cardiovascular disease guidelines recommend low-fat diets?;

This sentence has been changed according to reviewer comments. However, it should be noted that low-fat diets have been applied some decades before as the main strategy for people with obesity to reduce caloric intake. Current evidences from some clinical studies, such as the PREDIMED study, a nutritional intervention trial carried on people with high cardiovascular risk factors, reported no beneficial effect of low-fat diets contrary to Mediterranean diet supplemented with olive oil or nuts that reduced 30% the risk on new onset of cardiovascular event (Estruch R, Ros E, Salas-Salvadó J, Covas MI, Corella D, Arós F, Gómez-Gracia E, Ruiz-Gutiérrez V, Fiol M, Lapetra J, Lamuela-Raventos RM, Serra-Majem L, Pintó X, Basora J, Muñoz MA, Sorlí JV, Martínez JA, Fitó M, Gea A, Hernán MA, Martínez-González MA; PREDIMED Study Investigators. Primary Prevention of Cardiovascular Disease with a Mediterranean Diet Supplemented with Extra-Virgin Olive Oil or Nuts. N Engl J Med. 2018 Jun 21;378(25):e34.)

- Rows 52-55: “Other dietary plans have been proposed including low-carbohydrate diets, high-protein diets, very low-caloric diets with meal replacements, Mediterranean diet, and diets with intermittent energy restrictions, evidencing that a successful diet must be healthy,balanced and without nutritional deficiencies” – please rephrase for more clarity;

Sentence has been modified to clarify the meaning

- Rows 83-84: “giving relevance to the functional roles of the different fatty acid residues in lipids” – rephrase?

Sentence has been modified

- Regarding abbreviations, once they are presented in the text, please do not alternate abbreviation-full form; e.g. Row 205: “Linoleic acid” and “LA” in Row 214, or “Unsaturated Index” in Row 216, when the abbreviation should be used. Similarly, in sections 3.5 and 3.6 is it “FAs” or “FA-s”? Please double check and use the same abbreviation. Row 345: “D9D” – abbreviation not previously presented and in Row 348: “desaturase” - if you introduce the abbreviation delta-9-desaturase - D9D in the Materials and Methods section, you should further use it. If not, please replace D9D in line 345 and 354.

Corrections have been made

- Table 1: p=0.06 for “Stearic acid” – I think it would be interesting the state (and maybe discuss/look for other literature reports) that a borderline significance level was observed for stearic acid, values being higher for children (in opposition the palmitic acid) (?)

The most important results of this work are showed in Table 1 when we compare the RBC fatty acid profile between adults and children with normal weight or obesity. It is emerging the relevance of the palmitic acid level, and consequently of the decreased delta-9 desaturase transformation and unsaturation index, and the increased level of stearic acid (BORDERLINE) can also indicate the less efficiency of delta-9 desaturase also on the “main” transformation into oleic acid. This is not seen in the healthy groups. In children with obesity the desaturase enzyme is working less to create palmitoleic acid, which is an important adipokine.

- Could you motivate why you have chosen to assess the routine blood biochemical parameters only for a subsample of your study population? I find this to be a serious liability of your study.

Indeed, it is considered as one of the limitations of the study as it is described in the Discussion section, rows 383-385. Unfortunately, the analysis of all the blood biochemical study samples could not be carried out due to different setbacks. In any case, we consider it interesting to plot the results of a partial sample of all participants, even though we are aware that future work should be done in order to have more measurements.

- Table 2: I believe you should also analyze the obese vs normal weight for both children and adults. You should firstly emphasize the differences between normal and obese for each age group in your study and then obese-obese and normal-normal. For example, TC and HDL are higher in non-obese children, but LDL is lower in obese. Are the differences significant? Maybe these data should be taken into consideration when you discuss the results of your study.

Even if we also consider it interesting, the main goal of this study was to stablish the differences between children and adults with obesity to personalize better the intervention strategies. Even if the median of LDL Cholesterol is higher for children with normal weight, compared to children with obesity, after Mann-Whitney test was carried out with a non-significative result for LDL cholesterol (p=0.189) was observed. For Total Cholesterol and HDL cholesterol, after Mann-Whitney test was carried out, significative results were obtained for both (p<0.001 and p=0.005). It is noteworthy to mention that, even if statistical significative differences are observed between children with normal weight and obesity, all the results are within the optimal ranges.

Maybe with a higher sample size of the blood biochemical parameters, different studies would be worthy.

- A small note – I appreciate the thorough statistical analysis applied. Thanks

- Rows 243-244: Again, here it might be useful to compare the two children groups (obese and non-obese) and the two adults’ groups. Comparing the intake (g) of children and adults, when their caloric necessities are different seems pointless and the p-values presented in the S1 table are irrelevant, in my opinion, and you could directly present/discuss Table 3.

Indeed, even if the comparison of the dietary intake between both groups can have low interest as they have different caloric necessities, as we stated in manuscript (row 248-249), the nutritional survey is always necessary when populations are compared, in order to understand  the role of specific food intakes in the metabolism and also to consider these variables  as a covariate in the ANCOVA analysis, as explained in the Methods section, rows 177-185 so it is worth to representing it in a table format in the results section.

The objective of this study was to stablish the differences between children and adults with obesity in order to personalize better the intervention strategies and normoweight groups were used to determine if those differences were only attributable to age, as explained in Discusion section, row 300-308. The characterization of the differences between children with normoweight and obesity have been studied in other publications(Jauregibeitia IP, K.; Rica, I.; Tueros, I.; Velasco, O.; Grau, G.; Trebolazabala, N.; Castaño, L.; Larocca, A.V.; Ferreri, C.; Arranz, S. Fatty Acid Profile of Mature Red Blood Cell Membranes and Dietary Intake as a New Approach to Characterize Children with Overweight and Obesity. Nutrients. 2020;12) and same happens for differences between adults with normoweight and obesity (Genio G. Morbid Obesity is Associated to Altered Fatty Acid Profile of Erythrocyte Membranes.Journal of Diabetes & Metabolism. 2015;06;Sansone A, Tolika E, Louka M, Sunda V, Deplano S, Melchiorre M, et al. Hexadecenoic Fatty Acid Isomers in Human Blood Lipids and Their Relevance for the Interpretation of Lipidomic Profiles. PloS one. 2016;11(4).)

- Row 255: “the pediatric population showed higher intake of stearic acid” – Was this previously reported? Are other literature sources that state the same? Is there an explanation?

In table 3, the pediatric population showed higher intake of palmitic acid, not stearic acid. The nutritional survey is always necessary when populations are compared, in order to exclude or include the role of specific food intake. On the other hand, we use this dietary information as a covariate for the Ancova analysis, as it is explained in the Methods section, rows 177-185.

- Table 3 – formatting issue: in the table head row the font for “Mann-Whitney U test” seems to be different from the one of the previous two column heads.

You were right, already fixed.

- Row 297: “higher levels of w6/w3 ratio in RBCs for adults” - please double check and clarity, as in Table 1, the higher values are presented for obese children.

Authors have corrected this sentence in order to clarify that the higher levels of ω6/ω3, observed for adults with obesity, correspond with intake and not with RBC levels.  Concerning the dietary pattern of each group, the higher levels of ω6/ω3 intake for adults with obesity was an interesting result, showing its relationship with the quality and not with the quantity of lipids intake.”

- Rows 300-301: “have been compared with a population of adults and children with normal weight” – As stated above, firstly, you should check if RBC profiles differ in normal vs obese groups, not just obese vs obese and normal vs normal, or, please indicate some previous studies that have already reported and established such differences. Or… Please explain why you chose not to make the suggested comparisons.

As it has been cited before in previous answers to reviewer, other studies revealed this evidence.

Round 2

Reviewer 1 Report

The authors have adequately met my points of criticism.

Reviewer 3 Report

I have no further  question.

Reviewer 4 Report

I see that the authors adressed my concerns and the manuscript is much improved.